

# *Staphylococcus aureus* viewed from the perspective of 40,000+ genomes

Robert A. Petit III and Timothy D. Read

Department of Medicine, Division of Infectious Diseases, Emory University School of Medicine, Atlanta, GA, USA

## ABSTRACT

Low-cost Illumina sequencing of clinically-important bacterial pathogens has generated thousands of publicly available genomic datasets. Analyzing these genomes and extracting relevant information for each pathogen and the associated clinical phenotypes requires not only resources and bioinformatic skills but organism-specific knowledge. In light of these issues, we created Staphopia, an analysis pipeline, database and application programming interface, focused on *Staphylococcus aureus*, a common colonizer of humans and a major antibiotic-resistant pathogen responsible for a wide spectrum of hospital and community-associated infections. Written in Python, Staphopia's analysis pipeline consists of submodules running open-source tools. It accepts raw FASTQ reads as an input, which undergo quality control filtration, error correction and reduction to a maximum of approximately 100× chromosome coverage. This reduction significantly reduces total runtime without detrimentally affecting the results. The pipeline performs de novo assembly-based and mapping-based analysis. Automated gene calling and annotation is performed on the assembled contigs. Read-mapping is used to call variants (single nucleotide polymorphisms and insertion/deletions) against a reference *S. aureus* chromosome (N315, ST5). We ran the analysis pipeline on more than 43,000 *S. aureus* shotgun Illumina genome projects in the public European Nucleotide Archive database in November 2017. We found that only a quarter of known multi-locus sequence types (STs) were represented but the top 10 STs made up 70% of all genomes. methicillin-resistant *S. aureus* (MRSA) were 64% of all genomes. Using the Staphopia database we selected 380 high quality genomes deposited with good metadata, each from a different multi-locus ST, as a non-redundant diversity set for studying *S. aureus* evolution. In addition to answering basic science questions, Staphopia could serve as a potential platform for rapid clinical diagnostics of *S. aureus* isolates in the future. The system could also be adapted as a template for other organism-specific databases.

## INTRODUCTION

*Staphylococcus aureus* is a common and deadly bacterial pathogen that has been frequently investigated by whole genome sequencing over the last decade. It was the subject of arguably the first large scale bacterial genomic epidemiology study using Illumina sequencing technology (*Harris et al., 2010*). The cumulative number of Illumina shotgun

Corresponding author
Timothy D. Read, tread@emory.edu

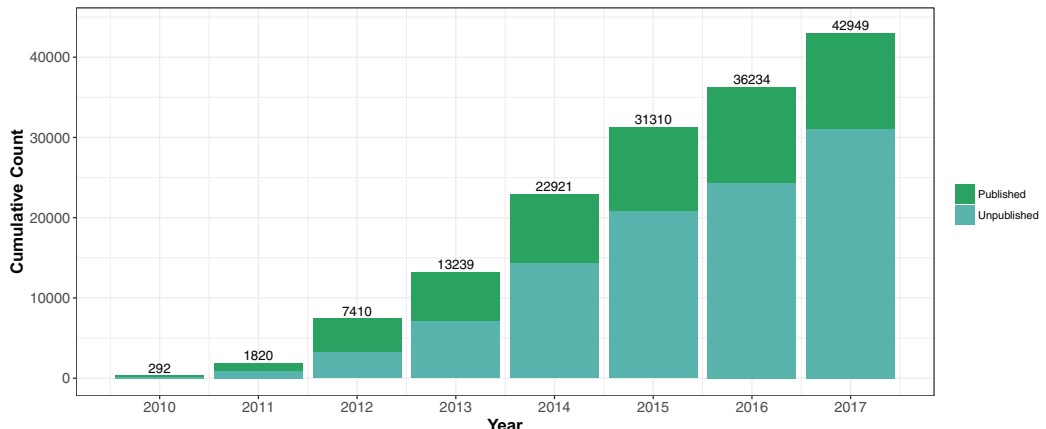

**Figure 1 Cumulative submissions of *Staphylococcus aureus* genome projects 2010–2017 linked to publications.** There were 42,949 *S. aureus* genome projects investigated in this study. Of these samples, we have linked 11,921 to a publication.

genome projects deposited in public repositories (the National Center for Biotechnology Information Short Read Archive (NCBI SRA) and the European Nucleotide Archive (ENA)) had grown to almost 50,000 by March 2018 (Fig. 1). *S. aureus* is therefore on the front edge of a cohort of bacterial species that are acquiring broad whole genome shotgun coverage, offering possibilities of new types of large scale analysis.

*Staphylococcus aureus* is a Gram-positive bacterium with a chromosome of ~2.8 Mbp. Plasmid content varies between strains. A multi-locus sequence typing (MLST) scheme that assigns each strain a "sequence type" (ST) based on seven genes has proven a robust way of describing individual strain genotypes and membership of larger "clonal complexes" (CCs) (*Planet et al., 2017*). The accumulated public *S. aureus* genome datasets present an opportunity for investigating basic questions about how genetic variations that cause antibiotic resistance evolve within populations and how long genes traded by horizontal gene transfer persist in populations. However, there has been a problem of access, as few published tools fill the niche of providing methods for fine-scale querying of very large datasets from a pathogen species. For example, PATRIC (*Wattam et al., 2014*) and BIGSdb (*Jolley & Maiden, 2010*) web based analysis sites focus on high quality annotation and core genome MLST (cgMLST), respectively, while Aureowiki (*Fuchs et al., 2017*) and PanX (*Ding, Baumdicker & Neher, 2018*) provide very detailed information on a smaller number of strains. In this study we describe the creation of Staphopia, an integrated analysis pipeline, database and application programming interface (API) to analyze *S. aureus* genomes.

# MATERIALS AND METHODS

## Staphopia analysis pipeline

The Staphopia analysis pipeline (StAP) processed FASTQ (pFASTQ) files from a single genome through quality control steps and bioinformatic analysis software. StAP (DOI 10.5281/zenodo.1255310) consisted of custom Python3 scripts and open source software organized by the Nextflow (*Di Tommaso et al., 2017*) (v0.28.2) workflow

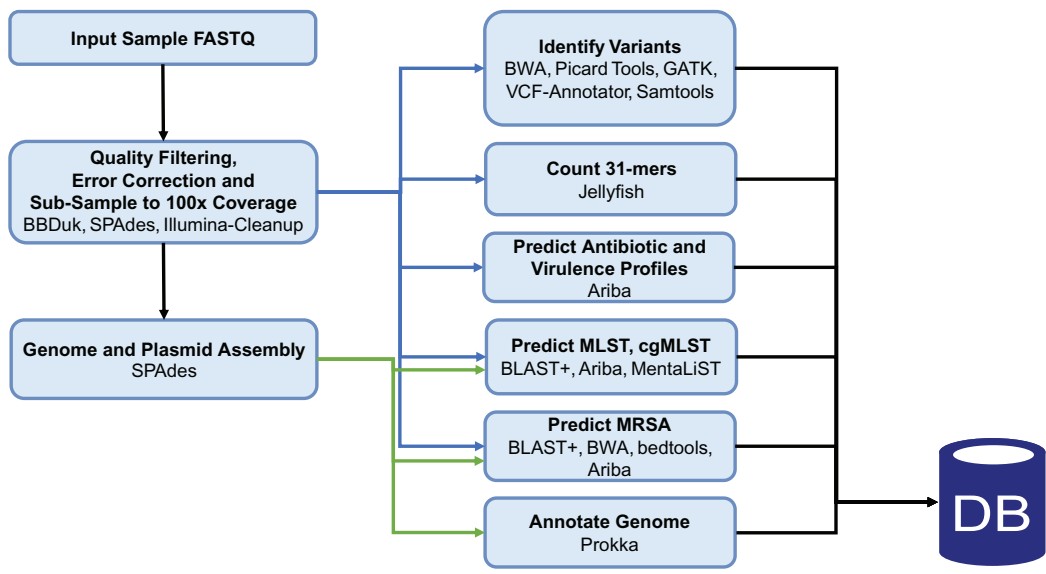

**Figure 2 Staphopia Analysis Pipeline (StAP) Workflow.** The diagram describes basic operations of the pipeline on a single genome input (FASTQ file) before uploading into the Postgres relational database. Details of the programs used are in the 'Methods' section and in DOI 10.5281/zenodo.1255310. Green arrows indicate input from de novo assembled contigs; blue arrows were operations performed on pFASTQ files.

management platform (Fig. 2). When available we used BioConda (*Grüning et al., 2017*) to install the open source software. Summary statistics of the original input and subsequent downstream results files were collected at each step of the pipeline. For portability, StAP was wrapped in a Docker container. The version of the pipeline used in this work was Docker Image Tag: 112017(https://hub.docker.com/r/rpetit3/staphopia/).

The input to StAP was either single- or paired-end FASTQ file (or files) generated from Illumina technology. StAP contained an option that allowed FASTQ data to be pulled from the ENA based on the experiment accession number (ena-dl v0.1, https://github.com/rpetit3/ena-dl). A MD5 hash (md5sum) was generated from the input FASTQ data and cross-referenced against a list generated from processed genomes to prevent reanalysis of the same input. BBduk (*Bushnell, 2016*) (v37.66) was used to filter out adapters associated with Illumina sequencing and trim reads based on quality. Read errors were corrected using SPAdes (*Bankevich et al., 2012*) (v3.11.1). Based on the corrected reads, low quality reads were filtered out and the total dataset was subsampled to a maximum of 281 Mbases (100× coverage of the N315 reference chromosome (*Kuroda et al., 2001*)) with Illumina-cleanup (v0.3, https://github.com/rpetit3/illumina-cleanup/). This file (or files, if paired-end) we termed "pFASTQ."

Processed FASTQ reads were assembled de novo using SPAdes (*Bankevich et al., 2012*) (v3.11.1). SPAdes also marked assembles as putative plasmids based on evidence such as relative read coverage (*Antipov et al., 2016*). Summary statistics of the assembly were created using the assembly-summary script (https://github.com/rpetit3/assembly-summary). A BLAST+ nucleotide database was created from the assembled contigs to be used

subsequently for sequence query matching. Open reading frames and their putative functions were predicted and annotated using Prokka (*Seemann, 2014*) (v1.12) and its default database.

The *S. aureus* strain N315 (*Kuroda et al., 2001*) chromosome (ST5 methicillin-resistant *S. aureus* (MRSA); accession NC_002745.2; length 2,814,816 bp) was used as a reference for calling consensus single nucleotide polymorphisms (SNPs) and indels in the pFASTQ reads using the GATK (*McKenna et al., 2010*) (v3.8.0) pipeline. GATK pipeline also incorporated BWA (*Li & Durbin, 2009*) (v0.7.17), SAMtools (*Li et al., 2009*) (v1.6) and Picard Tools (v2.14.1, http://broadinstitute.github.io/picard/) software. Identified variants were annotated using the vcf-annotator script (v0.4, https://github. com/rpetit3/vcf-annotator). Jellyfish (*Marçais & Kingsford, 2011*) (v2.2.6) was used to count *k*-mers of length 31 base pairs (31-mers) in the pFASTQ file. We used Ariba (*Hunt et al., 2017*) (v2.10.2) to make antibiotic resistance and virulence predictions for paired-end reads only. Resistance phenotypes were predicted using the MegaRes reference database (*Lakin et al., 2017*) and virulence using the Virulence Factor Database (*Chen et al., 2016*) core dataset.

Multi-locus sequence typing was determined by two or three methods depending on the whether the pFASTQ was paired- or single-end. All methods used the *S. aureus* MLST allele sequence database downloaded from https://pubmlst.org/saureus/ (November 2017). Alleles for each of the seven loci were aligned against the assembled genome using BLAST+ (v2.7.1+) (*Camacho et al., 2009*). Alleles and ST were determined based on perfect matches (100% nucleotide identity with no indels). We also used the MentaLiST (*Feijao et al., 2018*) (v0.1.3) software to call MLST based on *k*-mer matching of the alleles to the pFASTQ file. Unlike the BLAST+-based MLST method, MentaLiST did not require exact matches to alleles to predict a ST. If the pFASTQ was paired-end, Ariba (*Hunt et al., 2017*) (v2.10.2) also determined MLST alleles and ST. The default ST call for each genome was determined in the following order: agreement between each method, agreement between MentaLiST and Ariba, agreement between MentaLiST and BLAST+, agreement between Ariba and BLAST+, Ariba alone without a novel or uncertainty call, MentaLiST alone, and finally BLAST+ alone. cgMLST was determined with MentaLiST using the *S. aureus* cgMLST scheme (*Leopold et al., 2014*) available at http://www.cgmlst.org.

Evidence for Staphylococcal Cassette Chromosome mec (SCC*mec*) predictions were based on multiple approaches. The primary approach aligned SCCmec typing primers, downloaded from http://www.staphylococcus.net/, against the assembled genome using BLAST+ (v.2.7.1+) (*Camacho et al., 2009*). Samples with a perfect match to primer pairs for a given amplicon were assigned an SCCmec type following the *Kondo et al. (2007)* algorithm. Genes and proteins associated with SCCmec mere aligned against the assembled genome using BLASTN and TBLASTN. We also mapped the pFASTQ to each SCCmec cassette using BWA (*Li & Durbin, 2009*) (v0.7.17). The overall cassette and *mec* region coverage statistics were determined as well as the per-base coverage for each cassette using genomeCoverageBed (*Quinlan & Hall, 2010*) (v2.26.0). The methods described above were based on the 11 SCC*mec* types currently

listed in the http://www.sccmec.org (I–XI) and hence did not include recently described types XII and XIII (*Wu et al., 2015*; *Kaya et al., 2018*). We labelled a genome as "MRSA" only if each *mecA* typing primer (*Kondo et al., 2007*) had a perfect BLASTN match on the de novo assembly, a predicted *mecA* gene ortholog had a BLAST score ratio of at least 95%, or Ariba (*Hunt et al., 2017*), as previously described, predicted reads in the paired-end pFASTQ file matching a *mecA* target.

### Web application, Relational Database and application programming interface

We used Django (v2.0), a Python web framework, to develop a PostgreSQL (v10.1) backed relational database for storing the results from the analysis pipeline (Fig. S1). A Django application was created for each module of the pipeline, automating the creation of database tables for the results. Python scripts building off Django were developed for insertion of results from each StAP module or the StAP as a whole. A web front-end was developed (https://staphopia.emory.edu) using the Bootstrap (v4.0) and jQuery (v3.2.1) web frameworks. We used the Django REST framework to develop an extensive API that allowed users to create queries accessing multiple samples. We also developed an R package, Staphopia-R (DOI 10.5281/zenodo.1255314), to programmatically access the API. The API and its endpoints were documented to allow users to further develop their own packages in a language of their choice. The source code for our web application was made available at DOI 10.5281/zenodo.1255312.

### Processing public data

We used the Cancer Genomics Cloud (CGC) Platform, powered by Seven Bridges (http://www.cancergenomicscloud.org/), to process *S. aureus* genomes through StAP in November 2017. CGC allows users to create custom workflows based on Docker containers, then execute these workflows on the Amazon Web Services (AWS) cloud platform. We obtained a list of publicly available *S. aureus* sequencing projects from the ENA web API using the following search term:

> "*tax_tree(1280) AND library_source=GENOMIC AND (library_strategy=OTHER OR library_strategy=WGS OR library_strategy=WGA) AND (library_selection=MNase OR library_selection=RANDOM OR library_selection=unspecified OR library_selection="size fractionation")*."

We only processed projects which used Illumina sequencing technology. CGC opened AWS r3.xlarge instances (30.5 GB RAM, four processors) that downloaded FASTQ files from the ENA, using ena-dl for each genome, and ran the StAP pipeline. Ouput files were returned to the CGC, then uploaded into the Staphopia database server.

### Metadata collection

We used the ENA API to download and store any information linked to the "Experiment," "Study," "Run" and "BioSample" accessions into the database for each genome. We also determined each sample's publication status using three approaches.

The first approach identified existing links between SRA, a mirror of ENA, and PubMed using NCBI's Entrez Programming Utilities web API (*Entrez Programming Utilities Help, 2010*). For any links identified, we used the corresponding PubMed ID to extract information corresponding to the publication and stored them in the database.

The second approach searched for accessions within the text of scientific articles. We searched PubMed using the term, "*S. aureus,*" limited to the years between and including 2010 (the date of the first publicly available Illumina data upload), and 2017. The saved results, stored as XML, were then loaded into Paperpile, a subscription-based reference management tool, and the corresponding main-text PDFs were automatically downloaded. This process did not include supplementary information files, which required a manual operation. For those articles in which a PDF could not be automatically downloaded, attempts to manually acquire the PDF were made. Using the text search program "mdfind," available on Apple OS X, each accession (BioSample, Experiment, Study and Run) in the Staphopia database was used as a separate query to search all the PDF files. Experiment accessions with a corresponding PubMed ID were then stored in the database. In cases where a Study, BioSample or Run accession was identified in PDF text, each associated Experiment accession was linked to the corresponding PubMed ID.

For the third approach, a collection of PubMed articles with primary descriptions of *S. aureus* genome sequencing studies was manually curated. For these studies, the PDF and all available supplementary information were downloaded. The process of text-mining the articles and linking Experiment information to PubMed ID was repeated as described in the second approach. A list of these articles is available in BibTeX format from the code repository.

## Creating non-redundant *S. aureus* diversity set

Using available metadata, we selected a non-redundant diversity (NRD) set of genomes that were "Gold" quality (see "Results"), linked to a publication and each had a unique ST. When more than one strain from a ST was available, we randomly selected one individual giving priority to samples with collection date, site of isolation and location of isolation fields filled.

Using predicted variants against N315, we extracted a list of genes that had complete sequence coverage (i.e., "core" genes) but no predicted indels. For each reference gene sequence, we created an alternative gene sequence with SNPs predicted in each sample. If all the 31-mers of the alternative gene sequences were present in a 31-mer database of the pFASTQ file of genome created using the Jellyfish (*Marçais & Kingsford, 2011*) tool, the reconstructed sequence was considered "validated." Validated reconstructed gene sequences or all genomes were stored in the database and made available through the API for rapid phylogenetic comparisons.

A total of 380 genomes were selected for phylogenetic analysis as representatives of unique STs with good quality metadata. The set of validated genes were extracted and concatenated into a single sequence for each genome and saved in multi-FASTA and PHYLIP formats. A guide tree was generated with IQ-Tree (*Nguyen et al., 2015*)

(v8.2.11, -fast option) for identification of recombination events with ClonalFrameML (*Didelot & Wilson, 2015*) (v1.11). A recombination free alignment was created with maskrc-svg (https://github.com/kwongj/maskrc-svg). We used IQ-Tree to generate the final maximum likelihood tree with the GTR model and bootstrap support. Bootstrap support was generated from 1000 UFBoot2 (*Hoang et al., 2018*) (ultrafast bootstrap) replicates. We annotated the tree using iTOL (*Letunic & Bork, 2016*).

## RESULTS

### Design of the Staphopia analysis pipeline and processing 43,000+ genomes

The StAP (Fig. 2) was written to automate processing of individual *S. aureus* genomes from Illumina shotgun data. The pipeline was designed as a series of modules running individual software packages, organized by the Nextflow (*Di Tommaso et al., 2017*) workflow language, which made it possible to run the entire pipeline or individual components as needed. The first step of the pipeline was to import single- or paired-end FASTQ files either as local files, or from the ENA database. Following quality-based trimming and down selection of the FASTQ to 281 Mbases (~100× coverage of the N315 reference chromosome (*Kuroda et al., 2001*), NC_002745.2), analyses were run on the raw pFASTQ files directly, or on de novo genome assemblies constructed by the SPAdes program (see "Methods" for more details). We decided to downsample the input FASTQ files for two reasons: to manage the computational burden when running thousands of genome projects and also to achieve genome datasets with consistently sized pFASTQ input files. The threshold of ~100× coverage was chosen after preliminary studies (https://github.com/staphopia/staphopia-paper/blob/master/analysis/07-supplementary.pdf) showed that there was either small or no improvements in outcome for downstream assembly and remapping steps for input files >100× but large increases in processing time and memory requirement. We created a Postgres database to store results from the StAP analysis and a web front end and a web API for mining the data. An R package (Staphopia-R) was written for interacting with the API and was used for most analysis presented in the results.

In November 2017 there were 44,012 publicly-available shotgun sequencing projects with FASTQ files in ENA. Illumina technology was the dominant platform, accounting for 99% of samples ($N = 43,972$). A total of 81% ($N = 35,580$) of them had at least 281 Mbases sequence data. We processed all Illumina genomes in parallel through the StAP using cloud servers (see "Methods"). On parallel r3.xlarge instances with 30.5 Gb RAM and four processors, the mean time to process a genome was 52 min with an interquartile range of 47–56 min (Fig. S2).

### Sequence and assembly quality trends

We identified samples that were likely mixed-samples or not *S. aureus* whole genome shotgun projects and/or were of low technical quality and marked them to not be included in subsequent analysis. We removed genomes that failed to match to any known allele of the seven MLST loci (323 genomes), had a total assembly size that differed

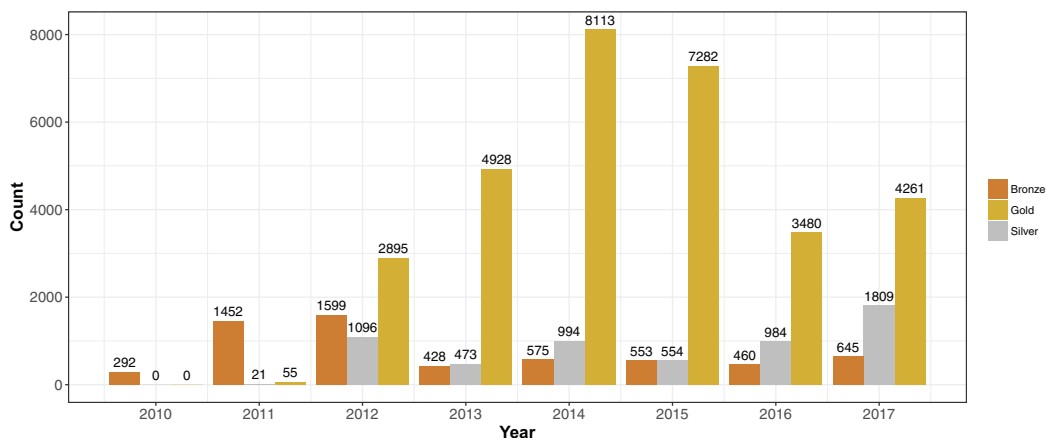

**Figure 3 Sequencing quality ranks per year 2010–2017.** Genome projects were grouped into three increasing quality ranks: Bronze, Silver and Gold. The rank was based on coverage, read length and per-read quality (please see "Results"). The highest rank, Gold, represented 72% ($N = 31{,}014$) of the available *S. aureus* genome projects. The remaining genomes were almost evenly split between Silver ($N = 5{,}931$) and Bronze ($N = 6{,}004$).

by more than one Mb from a typical *S. aureus* chromosome (<1.8Mb or >3.8Mb; 764 genomes), or had a GC content differing more than 5% (<28% or >38%; 467 genomes) of the expected 33% GC content. Failure to complete the StAP pipeline due to poor data quality, and coverages less than $20\times$ were flagged in 101 and 142 genomes, respectively. In total, we removed 1,023 genome projects, leaving 42,949 for further analysis.

We placed genomes into an arbitrary ranking of 1–3 ("Bronze," "Silver" and "Gold") based on the pFASTQ coverage and average sequencing quality. Paired-end genomes that had read lengths exceeding 100 bp, a coverage of $100\times$ and an average per base quality score of at least 30 were given a Gold rank. The purpose of the Gold rank was to group together high-quality samples with near-identical coverage. Paired-end genomes with similar read length and quality cutoffs but a lower sequence coverage (between $50\times$ and $100\times$) were classified as Silver. The remaining samples were given a rank of Bronze. Single-end reads were classified as Bronze no matter the read length, quality or coverage. More than 70% of the samples were of rank Gold ($N = 31{,}014$). There were 5,931 Silver and 6,004 Bronze rank samples. Each year since 2012, the number of Gold ranked genomes have exceeded Silver and Bronze (Fig. 3).

Changes in sequence quality and de novo genome assembly metrics over time reflected the development of Illumina technology. Mean per-base quality scores increased from ~32 in 2010 to >35 in 2012 and have stayed at that level since. The mean sequence read length rose in steps from <50 in 2010 to ~150 bp in 2017. Assembly metrics such as N50 (*Earl et al., 2011*), and mean and maximum contig length have gradually increased since 2010. Bronze ranked genome projects had similar (or sometimes even higher) mean per read quality scores than Gold and Silver since 2011. However, Silver and Gold assembly metrics such as N50 and mean contig size were generally quite similar and higher than Bronze.

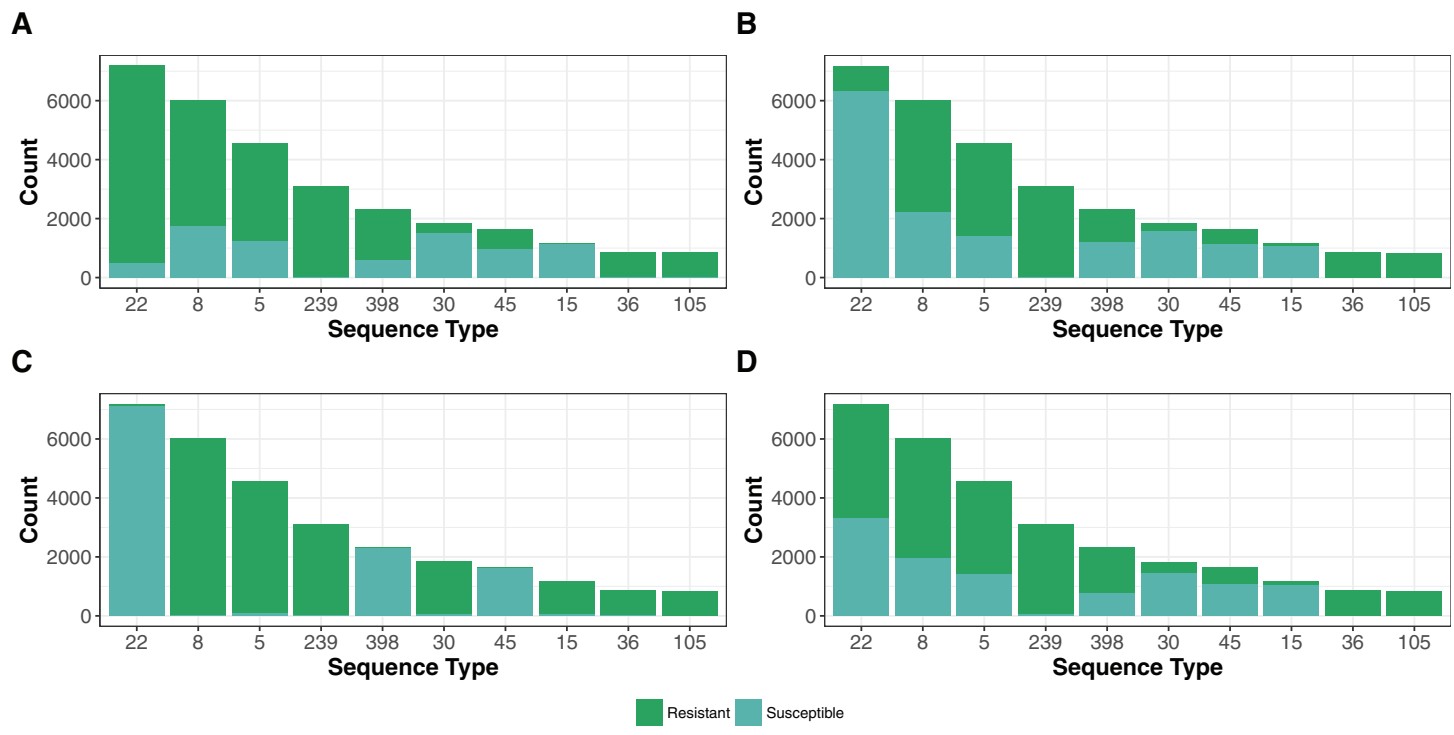

**Figure 4  Resistance genes to methicillin (MRSA), aminoglycoside, fosfomycin and macrolide-lincosamide-streptogramin (MLS) antibiotic in the top 10 STs.** The presence of resistance genes was predicted by Ariba (*Hunt et al., 2017*) using the reference MegaRes (*Lakin et al., 2017*) database. The distribution MegaRes resistance classes for methicillin (A), aminoglycosides (B), fosfomycin (C) and macrolide-lincosamide-streptogramin (D) are presented for the top 10 sequence types (ST). The top 10 STs represent 70% ($N = 29,851$) of the genomes analyzed in this study. A breakdown of each remaining resistance class tested is available in the code repository.           

## Genetic diversity measured by MLST

We obtained a view into the genetic diversity of the sequenced *S. aureus* genomes by in silico MLST using Ariba (*Hunt et al., 2017*), MentaLiST (*Feijao et al., 2018*) (both taking pFASTQ as input, but using different algorithms) and BLASTN against assembled contigs. A ST was assigned to 42,337 (98.6%) genomes. Of these, 41,226 (97.3%) calls were in agreement between MentaLiST, BLAST+ and (if paired-end) Ariba methods; 828 had agreement between two methods and a no-call on the other, and 189 were supported by one program with no-calls from the other two. Of the remaining 612 genomes not assigned to a known ST, 306 were predicted to be in a novel ST based on matches to known alleles of each of the seven loci. The remaining 306 genomes had 1–6 known *S. aureus* MLST alleles.

The 42,337 genomes assigned to existing STs represented only 1,090 STs of 4,466 in the PubMLST *S. aureus* database (November 2017, https://pubmlst.org/saureus/). The abundance distribution was weighted toward common strains, with the top 10 (STs 22, 8, 5, 239, 398, 30, 45, 15, 36 and 105) representing 70% ($N = 29,851$) of the genomes (Fig. 4).

The cgMLST set of 1861 loci (November 2017, https://www.cgmlst.org/) were assigned to the genome set using MentaLiST. There were 38,677 distinct patterns, with only

| SCCmec type | Count |
|---|---|
| Table 1 Predicted SCCmec cassette type representation. | |
| I | 689 |
| II | 5,183 |
| III | 2,807 |
| IV | 14,526 |
| V | 1,684 |
| VI | 171 |
| VII | 19 |
| VIII | 468 |
| IX | 0 |
| X | 20 |
| XI | 895 |

Note:
There were 26,462 samples with reads mapped to at least 50% of a SCCmec cassette. The table is a breakdown of the SCCmec cassettes with the highest percent match for each sample.

1,850 patterns found in more than one sample, the remaining 36,827 patterns were represented by a single genome.

## Antibiotic resistance genes

Treatment of *S. aureus* infections has been complicated by the evolution of strains resistant to many commonly used antibiotics (*Foster, 2017*). In particular, MRSA, carrying the *mecA* gene encoding the PBP2a protein that confers resistance to beta-lactam antibiotics, has become a global problem. We designated a genome as MRSA if each *mecA* typing primer (*Kondo et al., 2007*) had a perfect BLASTN match on the de novo assemblies (26,743 strains), a predicted *mecA* gene ortholog had a BLASTN score ratio of at least 95% (26,430 strains), or the Ariba (*Hunt et al., 2017*) algorithm predicted reads in the paired-end pFASTQ file matching a *mecA* target in the MegaRes (*Lakin et al., 2017*) database (27,120 strains). The number of genomes having at least one of these criteria (27,628) was 64% of the total number. Of these, 95% (26,340) of the samples had agreement between each of the criteria. The top five most common STs had a large portion of MRSA strains (Fig. 4), which reflects the selection bias of the research community in investigating these significant hospital and community pathogen strains over other *S. aureus*.

The *mecA* gene is usually horizontally acquired as part of a mobile genetic element called SCC*mec* (*Katayama, Ito & Hiramatsu, 2000*). SCC*mec* elements have been classified into at least 11 classes that vary in composition of *mec* genes, *ccr* cassette recombinase genes and spacer regions (http://www.sccmec.org). Knowledge of the SCC*mec* type can be useful for high-level characterization of MRSA strain types (*Kaya et al., 2018*). We showed that 10 of the 11 cassettes in the current schema map to at least one genome with highest coverage (an approximate method for assigning SCCmec type) (Table 1). Of the 26,462 (26,185 paired-end) genomes with at least 50% cassette coverage, 96%, 96% and 99% are MRSA based on primer BLASTN, protein BLASTN or MegaRes,

| Table 2 Antibiotic resistance classes predicted by non-core genes. | |
|---|---|
| **Antibiotic resistance class** | **Count** |
| Aminocoumarin | 46 |
| Aminoglycoside | 17,968 |
| Beta-lactam | 37,758 |
| Fluoroquinolone | 69 |
| Fosfomycin | 24,205 |
| Fusidic Acid | 346 |
| Glycopeptide | 5,777 |
| Lipopeptide | 44 |
| Macrolide-lincosamide-streptogramin (MLS) | 22,322 |
| Multi-drug resistance | 13,653 |
| Phenicol | 852 |
| Rifampin | 46 |
| Sulfonamide | 36 |
| Tetracycline | 8,638 |
| Trimethoprim | 6,605 |

Note:
Number of genomes with genes of resistance classes predicted by Ariba using the reference MegaRes database naming scheme.

respectively. All type XI cassettes were *mecA* negative by primer BLASTN because these contained the *mecC* allele (*García-Álvarez et al., 2011*; *Shore et al., 2011*), which was sufficiently different to be outside the normal distance for a positive match. Additionally, we found 53 genomes which matched to at least 50% of a SCCmec cassette but were not MRSA and had no reads mapping to the *mec* region of the cassette.

In addition to *mecA*, we found numerous other classes of non-core antibiotic resistance genes using the MegaRes (*Lakin et al., 2017*) class designations (Table 2). We did not consider SNPs/indels in core genes associated with resistance for this analysis. The most common class of resistance genes were beta-lactamases found in 37,758 genomes. Following this, the most common were the genes putatively conferring fosfomycin, macrolide-lincosamide-streptogramin and aminoglycosides resistance (24,205, 22,322, 17,968 genomes respectively). As with MRSA, the other common resistance genes were not distributed evenly among the top ST groups (Fig. 4), reflecting sampling ascertainment bias and also possibly differences in geographic distribution and prevalence of healthcare-isolated strains in the most common genotypes.

## Publication, metadata and strain geographic distribution

One challenge to using publicly available datasets through ENA or SRA is determining whether there is a published article describing the sequenced genome. We found through NCBI's Entrez Tools (eLink) that 6,712 genomes were linked to 48 publications in PubMed (March 2018). We attempted to add to the number by using text-mining methods to find *S. aureus* accession numbers in PDFs of *S. aureus* genome publications, ascertaining an additional 5,209 genomes in 30 publications. Therefore, of the 42,949

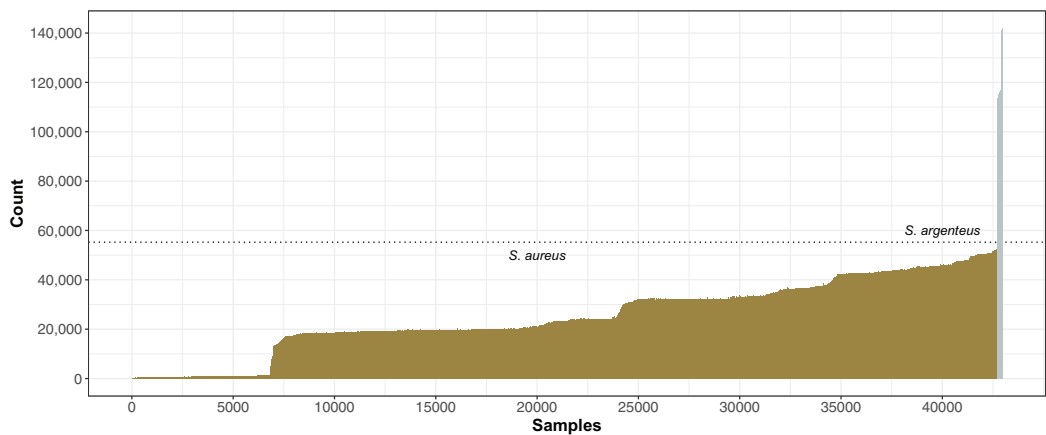

**Figure 5 *Staphylococcus aureus* SNP distance from reference *S. aureus* N315.** For each genome, the number of SNPs found by mapping reads to the N315 reference using GATK (*McKenna et al., 2010*) was plotted, with genomes ordered from least to most SNPs. A total of 240 genomes with >55,000 SNPs (dotted line) that had best matches to *S. argenteus* using mash (*Ondov et al., 2016*) were indicated by silver bars, the rest were *S. aureus* (gold).

samples deposited between 2010 and 2017, only 28% ($N = 11,921$) could be linked to a publication (Fig. 1). Since many genomes have been deposited in the last 1–3 years, this reflected the often significant time lag between depositing sequence data and final publication.

We noted that collection of metadata from public sequencing projects was another challenge. When submitting genome sequences to databases only a limited number of metadata fields are required, leading to the bulk of the information needing to be extracted manually from a publication, if it can be found. Only 40% ($N = 17,034$) genomes had a collection date, 35% ($N = 14,983$) had a geographic location and 35% ($N = 14,768$) had isolate source metadata. Using the available geographic data to geocode the sites of collection, we found that strains were from five continents and at least 40 countries. There was a strong bias toward strains from Europe ($N = 7,314$) and North America ($N = 5,882$), reflecting where the funding for most of the early sequencing studies had originated.

### A non-redundant *S. aureus* diversity set

The number of SNPs compared to the N315 reference strain varied from 6 to 141,893 within our collection of 42,949 genomes. The stepped pattern of the distribution (Fig. 5) reflected the organization of *S. aureus* into CCs. Apart from CC5 strains closely related to N315, the majority of *S. aureus* had ~50–50,000 SNPs and ~500–1500 indels called by the GATK pipeline (*McKenna et al., 2010*). There were a group of 240 most distant strains with >55,000 SNP (Fig. 5) that were found to be closer to the sister species, *S. argenteus* (*Holt et al., 2011*) based on ANI imputed by mash (*Ondov et al., 2016*), although 230 of these were assigned a *S. aureus* ST.

Of the 6,904 *S. aureus* genomes of Gold rank linked to a publication we selected a group of 380, each having a distinct ST as a NRD set of genomes. Of the 2,756 annotated N315 genes (excluding RNAs), 1,113 genes had no indels when reads from

each genome in the NRD dataset were mapped. Of these, 878 were "core" genes found in every genome. We reconstructed these genes for each of the NRD genomes starting with the N315 sequence and substituting predicted SNPs. These predicted sequences were then validated by decomposing into 31-mers and cross-checking whether each *k*-mer was present in pFASTQ files processed by Jellyfish (*Marçais & Kingsford, 2011*). We concatenated the 878 genes for each member of the NRD set and created a tree based on the 44,377 variant SNP positions (Fig. S3). The structure of the unrooted species tree resembles previous *S. aureus* phylogenies (*Planet et al., 2017*). The assembled genomes and concatenated gene sequences from the NRD dataset is available at DOI 10.6084/m9.figshare.6263435.

## DISCUSSION

The huge public library of genome sequence projects of *S. aureus* and other pathogens are a resource for microbiologists for testing genetic hypotheses in silico. Unfortunately, this has been a library of blank covers: most projects cannot be browsed to identify features such as underlying data quality, ST, key SNPs and non-core genes. Staphopia makes the library searchable for a number of important attributes, and we have described example workflows in the results section. Staphopia is unique in allowing both stand-alone analysis of *S. aureus* genomes through the StAP in a Docker container as well as access to the database of >40,000 processed genomes through more than 350 API endpoints. Furthermore, we have introduced a "Gold" rank for Illumina shotgun projects of 100× coverage and an average per base quality score of Q30 that allows comparison of more than 30,000 of the genomes with near-identical underlying data quality.

We used three strategies for analysis of raw sequence data: mapping reads to a reference chromosome to identify variants; de novo genome assembly, and direct analysis of the reads. Each has its strengths and weaknesses. Reference mapping retains quality information about variant calls but is limited to regions of the core genome and accuracy is reduced as genetic distance increases between the query and the reference. De novo assembly allows for discovery of novel accessory genes and is reference independent but could be affected by genomic contamination, and with Illumina short read data small portions of the sequence could be lost in gaps between contigs. Direct analysis of reads based on *k*-mer decomposition approaches allows examination of sequence independent of mapping and assembly algorithms but is also susceptible to false results arising from contamination and random sequence error. Using different approaches to cross-validate wherever possible builds confidence and we showed that MLST and MRSA/MSSA identification were robust with different underlying data types collected.

We showed that *S. aureus* genomes sequenced to date are heavily over-represented by a small number of common STs (Fig. 4), with a large tail of rarer strains. This is similar to what is seen in the PubMLST database, where the 10 most common STs were also the top 10 STs in Staphopia. It is tempting to assume that these numbers are a reflection of the true population structure of *S. aureus*, and that this is dominated by a relatively

small number of recently expanded lineages. However, strains selected for sequencing are far from a random sample of the population, and have strong biases in geography and toward those found in hospitals. For this reason, we selected a non-redundant set from within the Staphopia genomes for investigators to use for cross-species bioinformatic studies. A total of 3,371 of PubMed STs were found not represented in Staphopia. Of these STs, 98.5% were represented by one or two isolates in PubMLST. ST390, with 36 samples from Pleven, Bulgaria between 1995 and 1999, was the highest represented ST without assignment in Staphopia. The large number of rare PubMLST STs missing from Staphopia could be a combination of sequencing errors in MLST data creating spurious STs, systematic undersampling of rare and geographically diverse strains in early Illumina genome projects, or simply that the population structure of *S. aureus* has a large number of rare STs.

There are many possible avenues for future extensions of the project. New tools for efficient direct querying of raw reads have recently become available (e.g., BigSI (*Bradley et al., 2017*), and mash (*Ondov et al., 2016*)) and we plan to incorporate them in future iterations of the pipeline. Some of the principal improvements need to be in protein functional annotation. For speed and simplicity, we elected to map genes called from de novo assemblies against the included Prokka (*Seemann, 2014*) RefSeq database. This has the advantage of giving consistent proteins naming that can be linked to many functional annotation databases through UniProt cross-references. However, for fine resolution studies of sets of genomes from Staphopia, we recommend reprocessing with Roary (*Page et al., 2015*) to incorporate paralog detection and to use more extensive databases for homology matching. Even then, specific modules would need to be incorporated to improve naming of intrinsically hard to annotate protein families (e.g., microbial surface components recognizing adhesive matrix molecules (*Foster et al., 2014*)).

A key problem highlighted in this study is the difficulty in tracing publications linked to public genome data and finding typical metadata on strains (date and place of isolation, body site). We were able here to link thousands of records to publications through searching text in PDFs. For this reason, we urge researchers publishing microbial genomes to quote the Project ID (i.e., the PRJN ID) of publicly submitted data in the full text of the publication. Extracting metadata from publications presented a more complicated process. Metadata is often available as spreadsheets, documents or PDFs which are not easily parsed. We believe that journals need to start enforcing machine readable standards for metadata associated with deposited strains. The routine usage of BioSample ID (https://www.ncbi.nlm.nih.gov/books/NBK169436/), which links strains to genomic information, would be a major step forward.

Staphopia was designed with Illumina shotgun data in mind but increased use of alternative sequencing technologies in the future may necessitate new development. "Long read" technologies (e.g., PacBio, Oxford Nanopore) tend to have assemblies with fewer gaps, higher per base errors and lower coverage. A "gold standard" PacBio assembly will have a different quality profile to Illumina technology data (which itself is also evolving). Another challenge for automated assembly of public data will be to

identify projects sequenced with multiple technologies and assembled as hybrids (e.g., as demonstrated by the Unicycler tool (*Wick et al., 2017*)). To do this would mean altering the pipeline to perform hybrid assembly when experiments with multiple technologies are associated with a strain. Currently, within ENA (and SRA) a BioSample can be associated with multiple Experiments, but an Experiment can only be associated with a single BioSample. When a BioSample was linked to more than one Experiment, it was difficult to determine in an automated way if it is actually the same genomic DNA input to multiple experiments or, in rare cases, a mistaken assignment of a set of genetically non-identical isolates with the BioSample (e.g., all isolates from a study given the common strain name "USA300"). Because of this, Staphopia treated each ENA Experiment as a unique sample, rather than the BioSample.

It is unclear at this time whether the approach of processing of every public dataset will be sustainable as sequencing data production grows in the future. It would only be possible if storage and processing costs fall faster than the accumulation of new data, and multi-genome database queries may still be prohibitively slow. An alternative strategy to processing all strains, would be to filter the isolates for redundancy, by removing isolates that are less than $n$ SNPs from any member of a canonical genome set. However, there is still information in deep sequencing studies that can be captured from distributions of reads and $k$-mer distribution, even if the consensus sequences of the strains are identical. The plasmid copy number may differ between clones grown under different conditions, and the distribution of reads across the genome can itself be used to infer relative growth rate (*Brown et al., 2016*). No two shotgun genome sequencing projects are identical, and all have some potential value, especially if they have strong supporting metadata.

## CONCLUSION

- We analyzed 43,972 *S. aureus* public Illumina genome projects using the newly developed "Staphopia" analysis pipeline and database. 42,949 genomes were retained for subsequent analysis after filtering against low quality.
- The data quality was high overall: 36,945 (86%) were from paired-end projects with greater than 50-fold coverage and 30 average base quality ("Gold" and "Silver" quality).
- There has been a great concentration of effort on a sequencing a small number of STs: only 1,090 STs of 4,466 previously collected STs were recovered and 10 STs make up 70% of all genomes.
- 26,340–27,628 genomes were predicted MRSA depending on the criteria used for classification.
- We could link only 28% of the genomes to a PubMed referenced publication.
- We identified 380 non-redundant highly quality published genomes as a reference subset for diversity within the species.
- We identified 878 core genes that can be reliably used for rapid tree building based on SNPs compared to the reference N315 genome.

## ACKNOWLEDGEMENTS

We would like to thank Tauqeer Alam, Jim Hogan, Santiago Castillo-Ramírez. Michelle Su, Michael Frisch and Erik Lehnert for their helpful suggestions. We would also like to acknowledge our gratitude to the many scientists and their funders who provided genome sequences to the public domain, ENA and SRA for storing and organizing the data, and the authors of the open source software tools and databases used in this work.

### Funding

Funding was from Emory University, Amazon AWS in Education Grant Program, and NIH grants AI091827 and AI121860. The Seven Bridges NCI Cancer Genomics Cloud pilot was supported in part by the funds from the National Cancer Institute, National Institutes of Health, Department of Health and Human Services, under Contract No. HHSN261201400008C. The funders had no role in study design, data collection and analysis, decision to publish, or preparation of the manuscript.

### Grant Disclosures

The following grant information was disclosed by the authors:
Emory University, Amazon AWS in Education Grant Program, and NIH: grants AI091827 and AI121860.
National Cancer Institute, National Institutes of Health, Department of Health and Human Services: under Contract No. HHSN261201400008C.

### Competing Interests

Timothy D. Read is an Academic Editor for PeerJ.

### Author Contributions

- Robert A. Petit III conceived and designed the experiments, performed the experiments, analyzed the data, contributed reagents/materials/analysis tools, prepared figures and/or tables, authored or reviewed drafts of the paper, approved the final draft.
- Timothy D. Read conceived and designed the experiments, performed the experiments, analyzed the data, contributed reagents/materials/analysis tools, prepared figures and/or tables, authored or reviewed drafts of the paper, approved the final draft.

### Data Availability

  Code for most analysis described in the Results section—DOI 10.5281/zenodo.1296448,
  Staphopia—https://staphopia.emory.edu,
  R Package—DOI 10.5281/zenodo.1255314,
  StAP—DOI 10.5281/zenodo.1255310,
  Web Package—DOI 10.5281/zenodo.1255312,

Docker Image—https://hub.docker.com/r/rpetit3/staphopia/,
NRD Dataset—DOI 10.6084/m9.figshare.6263435.

## Supplemental Information

Supplemental information for this article can be found online at http://dx.doi.org/
10.7717/peerj.5261#supplemental-information.

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
