# Peer review of "Staphylococcus aureus viewed from the perspective of 40,000+ genomes"

_PeerJ, doi:10.7717/peerj.5261_

## Round 0.1 · original submission · Minor Revisions

Both reviewers appreciated your attention to an important subject and were supportive of your manuscript. Please address all issues raised by each reviewer. In particular take note of Reviewer One's comment regarding articulating the value proposition for Staphopia when there are existing genome analysis/DB platforms (e.g. https://pubmlst.org/saureus/, http://www.irida.ca). I'd also ask you to specifically address the issue of future support for Staphopia - a perennial issue in this space.

·

Basic reporting

The PeerJ has a strange web form I have to fill out, which is expecting every manuscript to be a scientific paper with clear aims > hypotheses > results > interpretation. This doesn’t make any sense in this case, as much of this manuscript is actually describing a web resource. There isn’t an experiment per se. It has also asked me to rank comments in order of priority (which I have ignored and opted to go line by line). The most important issues for me have " ***** " around them. Should be easy to find. Anyway, I will do what I can.

This manuscript presents Staphopia, an online web resource/ database for genotyping data of s. aureus. Data has been gleaned from ~ 40K sequencing datasets from public database and seems to be primarily available through an API.

First of all, I will applaud the authors for wrangling such a large dataset. There is an incredible amount of logistics involved with serving this kind of data. There are some interesting ideas presented in the manuscript that I hope the authors can extend.

Line 18: Staphopia is such a catchy name.

*****Line 318. Are the data generated by staphoia submitted back to the main MLST database. This would be very useful. The true power of MLST comes from the standardisation and simple nomenclature it facilitates. (the same issue applies to the cgMLST data).*****


Line 58: “seven core genes alleles” doesn’t make any sense. Perhaps simply ‘seven genes?

Line 66: The authors may enjoy reading our paper: https://doi.org/10.1371/journal.pgen.1007261

*****Line 66: Much of the functionality presented in staphopia is already available through BigsDB. BigsDB is open source, and has been used by third parties to serve genomic data (SNPs, cgMLST, MLST) e.g. Insitute Pasteur has used it for a number of organisms e.g. http://bigsdb.pasteur.fr/klebsiella/klebsiella.html . BigsDB already has a S. aureus MLST database (https://pubmlst.org/saureus/ ), which is looking for curators. Perhaps something more like IRIDA http://www.irida.ca , could be used if BigsDB doesn’t suit. Anyway, this introduction gives a compelling argument for accessible web resources but fails to give the impetus for staphopia specifically. It is unclear how long the code base for staphopia will be supported for; many bioinformatic resources simply disappear as developers lose interest or funding. *****

Line 106: Prokka is usually rendered in title-case ‘Prokka’, not PROKKA. See Seemann et al bioinformatics 2014.

*****Line 139: there are a lot of citations here. There is no indications how these citations contribute to the SCCmec typing employed. The first one is for BLAST. The rest seem to be papers that described the primers? This should be separated out. Are all these citations necessary? Are all these primers presented in Kondo et al? Aren’t the sccmec types in sccmec.org, can you point to that ?*****

Line 139: This error occurs all through the manuscript. The authors have used BLAST+ (im guessing from the version number cited). They should cite https://doi.org/10.1186/1471-2105-10-421 . They should refer to BLAST as BLAST+ . There is a dramatic difference between BLAST from the 1990 Altschul paper and from the 2009 BLAST+ reboot.

Line 144: The English here is completely garbled and needs to be rewritten. “We also aligned proteins associated with SCCmec are also aligned against the assembled genome…the pFASTQ BWA” ??

Line 146: typo “to to” . Please review lines 137 – 152. There may be other errors.

Line 155: Supp Fig 1. The width of this figure is huge! And Far too detailed! This might be better in documentation, and a simpler schematic could be presented here for a general audience.

*****Line 170: Is this an ongoing agreement to use the Cancer genomic cloud? This has been used to generate data up to November 2017? But what about after this date? Does Staphopia keep up to date with new sequence data? On the basis of lines 169 – 185, it seems that this is a one off. This would drastically lower the utility of this service if this was the case.*****

Line 207: The use of numbering #1, #2 #3 with no paragraph breaks is very confusing. A figure of a workflow would be useful? Or at least some line breaks in the text.

Line 210: This gist might be better off in the main github. There is value in Keeping everything in one place.

Line 216: The number of STs / genomes are not mentioned.

Line 271: Does this really require a figure? The ranges are enough in my opinion.

*****Line 323: I am outraged with the presentation of the cgMLST data in this manuscript. Firstly and most importantly, there is NO CITATION OR DESCRIPTION for the scheme that was used. Based on the number of loci reported, I assume the authors have used the scheme described in Leopold et al, J. Clin. Microbiol. 2014, 52. This scheme is available at http://www.cgmlst.org , which should also be mentioned. According to the manuscript, Sequence types (cgMLST) were calculated for all genomes in staphopia. But the resulting data is does not appear to be available (no mention in the API documentation). Furthermore the final result in the manuscript simply states the number of resulting sequence types based on cgMLST and remarks that most genomes had their own unique cgMLST profile. This finding has already been presented many times before for cgMLST (see https://www.nature.com/articles/nmicrobiol2016185 for a good example). *****

Line 404-415: This appears to be methods. What does the Tree actually tell us?

Line 459: Typos. Project ID (i.e. the PRJN ID)

Line 460: ‘was much more manual’ ; clumsy.

Line 463: typo ID.

*****Line 484: There is no description of staphopia’s runtime and performance as it currently stands. *****

Line 488: We, along with many others, have found MLST, rMLST, cgMLST fairly useful for global, large scale comparisons. We only switch to SNPs for fine analysis (outbreak, clonal complex etc). I am surprised this not presented as an alternative here.

Line 501: clarify average quality (i.e. refer to average ‘base calling’ quality)

Line 512: Typo Cores.

*****Line 510: It would be good if these genomes were presented (as a table; genomes name, description, accession code) somewhere in the manuscript or website. *****

*****Line 512: It would be good if these genes were presented (as a table; gene name, description, locus tag) somewhere in the manuscript or website.*****

Line 590: Please italicize all binomials names in the references. Some gene names e.g. mec also need to be italics. As a tip, if youre using Endnote or something like that you can change the text formatting for stuff like this in the citation manager itself. Don’t try to convert the citations to plain text and then edit them. Trust me.

Line 675 & 696: Some citations have title case. Where Words Have An Upper Case Letter. Some do not. Please check journal style and make consistent. Again, make the changes in your citation manager.

Figure 5: You can actually infer everything in this figure from Figure 6. This figure is redundant.

Figure 6 & 7: I don’t think these need entire separate figure. It’s the same X axis just different Y. These might be easier as a composite figure where each graph is a subfigure (A, B, C…).

Figure 1: You can actually infer everything in this figure from Figure 8. This figure is redundant.

Figure 1: Why is 2018 included here? It’s only April. The graph implies only ~200 S. aureus were sequenced for all of 2018.

Figure legends: Some of these are very short. “Top ten STs” for figure 5 doesn’t explain anything. Consider that a lot of readers will go straight to the Figures and ignore the body text as first. Give them all the information they need to interpret the figure!

Experimental design

Line 80: I am impressed with the clarity of the methods section. Congratulations on publishing the pipeline on docker and the use of github. This is a step I wish more people would adopt.

Line 93: I agree with the use of subsampling here. citation?

Line 95: I am not a Staph person. Perhaps this is a well know type strain, but there is no explanation in this manuscript why N315 was chosen as the reference genomes for all read mapping and QC.

***** Line 108: I am unsure about whether using a single reference genome will suffice. Often using a single reference will introduce a major bias. Generally, genomes more distantly related to the reference have less SNPs called simply because less of the reference genome can be mapped or aligned. Perhaps S. aureus is uniform enough to permit this but for most species a single reference genome is a bad idea. Some effort has been made to control for this in the tree presented in Fig 10. But I do not believe this has been handled for general SNP data available through Staphopia. *****

Line 220: In my experience, people are fairly clueless when it comes to APIs. I would suggest includes a simple static download (csv, Excel) of this representative genome set. Along with a tarball of the assemblies. This will greatly improve the usage of this service.

Line 231: The tree construction method seems sensible.

Line 260: Is there a citation for downsampling reads in this way?

*****Line 274: This is an interesting approach. Using MLST to determine whether something is S. aureus or not. However, this introduces circular logic in next section, where the authors talk about the genetic diversity using MLST. They should have already thrown out any divergent S. aureus by filtering out these untypable strains, so theres no surprise that the vast majority of genomes in the next section (line 306) are typeable standard STs. If those 232 genomes aren’t S. aureus, then what are they? We’ve used Kraken (just on the assembled contigs) to search for contamination. *****

Line 333: The description of the method here seems very different to the methods section. The only citation is Larkin and Kondo. (see my previous point).

Validity of the findings

*****Line 167: According to the API documentation there is no way to search the entire database for a specific metadata field. There appear to be predefined search results (e.g all published strains, or all strains from 1/ST representative set) but all other access points seem to require a strain ID (which is specific to staphopia. How do I discover the data I want when I do not know the strain ID?

For instance, I want to fetch all strains from a particular BioProject (PRJEB2756). This is often the only accession number given in a publication. I would need to go through the metadata of the whole database i.e.
GET https://staphopia.emory.edu/api/sample/
Then I would need to grab all the strain IDs and then request again for the metadata. i.e.
POST https://staphopia.emory.edu/api/metadata/bulk_by_sample/
I do not know how many ID I could get at a time but the likelihood is that all the IDs at once (40K) is not possible. But OK, perhaps I pull down a few hundred at a time, over and over again. For each set I’ve pulled down, I would then need to iterate through each record looking for the BioProject of my choice. *****

Line 202: I am impressed with the effort to locate publications for the sequencing data.


Line 281: There is no comment on mixed samples. We see about 3-5% in the data we’ve looked at.

***** Line 311: This is deeply distressing that the MLST calling methods are not consistent. In a recent overview we determined Ariba was one of the best MLST callers but here there is a number of discrepancies. Which method is at fault? Is there some of systematic problem that is causing this issue? *****

Line 320: Is there an indication as to why so few STs have been sequenced? Many of these STs are probably errors (someone messed up their ABI traces), some of them might be very rare STs.

Line 339: 5 % of genomes had inconsistencies with calling mecA. Again, is this an error with one of the methods? Is one method suffering from some kind of limitation?

Line 371: I like this. We need good metadata and links to publications to understand the sequencing data. Otherwise the sequencing data is completely useless. This work here shows how poorly we, as a community, are with being upfront about our science.

Additional comments

*****Other: I am unsure of the scalability of the API given that it is serving large chunky files like genome assemblies as JSON. I ran 10 requests in parallel and found the turn around time was 20secs for one genome assembly. I’m going to guess that on a busy day with multiple users something like 1,000 genomes would take 5.5 hours to download. The best time I could get was about ~2sec / assembly. For someone to download all the assemblies (and people will try to do that), with no one else using the service, it will take about 22 hours. I wish I could suggest a solution, but this is a very real scalability problem. *****

Other: https://staphopia.emory.edu/api/sample/unique_st (no slash at the end) gives no response. I would expect an error message.

Other: The link for MIMS on the frontpage is broken: http://gensc.org/gc_wiki/MIGS/MIMS

Reviewer 2 ·

Basic reporting

The reviewed manuscript describes an integrated analysis pipeline, database and Application programming Interface that aims to facilitate large scale genomic analysis of the thousands S. aureus genomes that are publicly available. The authors assessed the quality of the sequenced sample and provide a curated list of 380 high quality genomes representative of S. aureus genetic diversity. The API, code and database made publicly available will form valuable tools for researcher interested S. aureus and microbial genomics. The manuscript is clearly written in unambiguous language with an appropriate structure and figures. The introduction provide sufficient background informations and relevant prior literature is appropriately referenced.

Experimental design

The work presented fits in the Biological sciences, Medical Sciences, and Health Sciences scope of the PeerJ journal. The research presented compile new informations on sequenced S. aureus isolates and provide valuable tool to mine new knowledge from thousands of available genome sequences. The method described are rigorous and accurately described. All the code necessary to replicate the findings is publicly available.

Validity of the findings

The analysis presented here shows that most of the misidentified S. aureus samples (corresponding to S. argenteus) still have an MLST allele in the saureus.mlst.net database. This strongly suggests that the total number of ST in saureus.mlst.net database is overestimated due to S. aureus misidentification. This comment doesn’t invalidate the conclusion statement of “a great concentration of effort on sequencing a small number of STs” as “10 Sts make up 70% of all genomes” but the statement “only 1,090 STs of 4,466 previously collected STs” (Line 505) and "only a quarter of ST were represented" (L34) should be tempered or at least discussed in the result section.

Additional comments

Minor comments and suggestions to improve the manuscript:

Line 31: Could you specify in the abstract which ST the chosen reference strain belongs to?

Line 92: What about non illumina-reads samples, were they incorporated into Staphopia database?

Line 106: Would be good to use a report annotations from one “high quality” (manually curated) S. aureus genomes. This would improve annotation consistency between the different genomes.

Line 117: Does it mean that if the pFASTQ file was single end no antibiotic/virulence prediction were done? Please clarify.

line 123: “for each” is repeated twice.

line 144-148: Typing errors, please rephrase

line 217: Could the authors specify how gold silver and bronze genome quality were defined (or refers to the result section)?

Line 231: Could the authors specify how many 31-mer validated genes were kept for downstream analysis?

Line 271: I don’t think that the analysis of the 44,012 genomes took more than 4 years (44,012 * 52min)? Could the author clarify this point?

Line 323: Providing access to the 1861 loci used for cgMLST will be valuable for the research community. Could the authors provide the coordinate of these loci?

Line 413: The number of variant SNP positions is inconsistent with the figure 10 legend (60,191 versus 44,377).

Figure 7: The legend of the bottom plot (aminoglycosides) is missing.

Line 423: Would be good to clearly state which important attributes are searchable in Staphopia.

Line 502: Is the threshold for gold and silver quality “35 average quality score” or 30 as specified in method line 287?

---

## Round 0.2 · accepted · Accept

Thank you for addressing all editor and reviewer comments. Please make sure you carefully review the proofs before publication as I noticed a few minor issues to address, eg line 237, spell out the number '380'.

#